# TRPA1 Polymorphisms Modify the Hypotensive Responses to Propofol with No Change in Nitrite or Nitrate Levels

**Isabela Borges de Melo [1], Gustavo H. Oliveira-Paula [2,†], Letícia Perticarrara Ferezin [3], Graziele C. Ferreira [2], Lucas C. Pinheiro [2], Jose E. Tanus-Santos [2], Luis V. Garcia [1], Riccardo Lacchini [3] and Waynice N. Paula-Garcia [1,*]**

[1] Department of Orthopedics and Anesthesiology, Ribeirao Preto Medical School, University of Sao Paulo, Ribeirao Preto 14048900, SP, Brazil
[2] Department of Pharmacology, Ribeirao Preto Medical School, University of Sao Paulo, Ribeirao Preto 14048900, SP, Brazil
[3] Department of Psychiatric Nursing and Human Sciences, Ribeirao Preto College of Nursing, University of Sao Paulo, Ribeirao Preto 14048900, SP, Brazil
* Correspondence: wgarcia@fmrp.usp.br; Tel.: +55-16-3602-2814
† Current address: Department of Medicine, Division of Cardiology, Wilf Family Cardiovascular Research Institute, Albert Einstein College of Medicine, New York, NY 10461, USA.

**Abstract:** Anesthesia with propofol is frequently associated with hypotension. The *TRPA1* gene contributes to the vasodilator effect of propofol. Hypotension is crucial for anesthesiologists because it is deleterious in the perioperative period. We tested whether the *TRPA1* gene polymorphisms or haplotypes interfere with the hypotensive responses to propofol. PCR-determined genotypes and haplotype frequencies were estimated. Nitrite, nitrates, and NOx levels were measured. Propofol induced a more expressive lowering of the blood pressure (BP) without changing nitrite or nitrate levels in patients carrying CG+GG genotypes for the rs16937976 TRPA1 polymorphism and AG+AA genotypes for the rs13218757 TRPA1 polymorphism. The CGA haplotype presented the most remarkable drop in BP. Heart rate values were not impacted. The present exploratory analysis suggests that TRPA1 genotypes and haplotypes influence the hypotensive responses to propofol. The mechanisms involved are probably other than those related to NO bioavailability. With better genetic knowledge, planning anesthesia with fewer side effects may be possible.

**Keywords:** propofol; TRPA1; adverse drug reactions; blood pressure; genetic polymorphisms; nitric oxide

## 1. Introduction

Propofol (2,6-diisopropylphenol) is a general anesthetic widely used for the induction and maintenance of anesthesia in different clinical scenarios [1,2]. Given its favorable pharmacokinetic and pharmacodynamic profile, including fast onset and offset effects as well as the low incidence of vomiting or nausea, propofol has been the most frequently administered anesthetic drug for the last 30 years [3]. Despite these advantages, propofol administration is commonly accompanied by exaggerated vascular relaxation [4–6]. These hypotensive side effects may lead to undesired cardiovascular effects, increasing morbidity and mortality risk [7–9].

Concerning Transient receptor potential ankyrin subtype 1 (TRPA1), its gene is in chromosome 8 in humans, and it is a ligand-gated cation channel that plays a pivotal role in regulating pain and inflammatory pathways [10]. Studies have discovered a potential anesthetic binding site in TRPA1 [11,12]. Notably, pungent anesthetics also produce an additional inhibitory effect besides activating TRPA1 [13,14]. It is unclear whether these effects are restricted to noxious anesthetics. A recent study [15] concluded that the effects are species-specific. Propofol inhibits rodent but not human TRPA1, and this anesthetic inhibits

TRPA1 by interacting at a distinct site from the activation site. There is evidence that propofol activates TRPA1 in sensory neurons and heterologous expression systems [13,14,16] and generally induces a paradoxical activation and sensitization of TRPA1 [17,18].

In addition to being expressed in sensory nerves, activation of vascular TRPA1 and the subsequent production of nitric oxide contributes to the vasorelaxation effects of propofol [19]. Moreover, recent evidence demonstrates a prominent role of TRPA1 in vasculature. Recent reports have confirmed that the TRPA1 channel regulates vasomotor tone [20–22]. Sinha et al. [19] demonstrate that propofol-induced depressor responses in vivo are predominantly mediated by TRPA1 ion channels and include activation of both NOS and BKCa channels. Later, Jhun et al. demonstrated that *TRPA1* gene polymorphisms and haplotypes contributed to the heterogeneity of acute pain [23].

However, the effects of TRPA1 polymorphisms on blood pressure responses to propofol have not yet been evaluated. Therefore, we hypothesized that TRPA1 polymorphisms might somehow correlate to the effects of propofol on blood pressure. The TRPA1 polymorphisms studied (rs920829, rs16937976 and rs13268757) were selected based on their prevalence and functional or clinical implications. They were associated with increased TRPA1 receptor activation in response to agonists and asthma in children [24]. TRPA1 rs920829, in addition to being correlated with asthma, also has an association with neuropathic pain and pain associated with sickle cell anemia [25]. This study tested whether the alterations in blood pressure and NO levels induced by propofol are affected by the genotypes for single nucleotide polymorphisms (SNPs) of TRPA1 (rs920829, rs16937976, and rs13218757) or their combination into haplotypes.

## 2. Materials and Methods

### 2.1. Subjects and Study Design

Patients undergoing colonoscopy (*n* = 164) were selected for the study according to the eligibility criteria, which included American Society of Anesthesiologists (ASA) physical status I or II, age between 20 and 80 years, and body mass index $\leq$30 kg/m$^2$. Patients with evidence of severe hypertension; uncontrolled respiratory, renal, hepatic, or hematological disorders; and a history of stroke or myocardial infarction were excluded from the study. We also excluded subjects with hypotensive episodes that required management with vasopressor drugs. All patients underwent bowel preparation within 24 h of the procedure according to the center's standard protocol, and the clinicians involved in data collection were blinded to the polymorphisms of interest.

After the initial evaluation, a venous blood sample was collected (at baseline). Then, propofol at a 2 mg/kg dosage was administered intravenously by bolus injection. A second blood sample was collected 10 min later. We registered systolic (SBP), diastolic (DBP), and mean (MBP) blood pressure values at baseline, and at 4 and 10 min after propofol infusion. Blood samples were promptly centrifuged at 1000× *g* for 3 min, and plasma aliquots were stored at −70 °C until analysis. The colonoscopy exam was initiated after the second blood sample was collected, preventing altered blood pressure levels or NO markers from being caused by the stress of the procedure. All patients were constantly monitored with cardioscopy (DII and V5), heart rate, non-invasive blood pressure, and blood oxygen saturation until complete recovery.

Exploratory endpoints were blood pressure changes calculated by subtracting the values observed 4 min after propofol anesthesia from the baseline values and NO marker delta levels calculated by subtracting the values observed 10 min after propofol anesthesia from the baseline values.

### 2.2. Measurement of Plasma Nitrite and Plasma Nitrate Concentrations

We measured plasma nitrite levels using an ozone-based reductive chemiluminescence assay [26], and the Griess reaction measured plasma NOx [nitrite + nitrate] concentrations [27]. Nitrate concentrations were calculated by subtracting the nitrate concentra-

tions as measured by chemiluminescence from plasma NOx levels as measured by the Griess reaction.

### 2.3. Genotyping

Genotypes for the TRPA1 SNPs were determined using TaqMan Allele TaqMan PCR. Thermal cycling was performed in standard conditions, and fluorescence was recorded by StepOne Plus Real-Time PCR equipment (Applied Biosystems, Foster City, CA, USA). Quality control of genotyping quality was carried out using previously assessed positive and negative controls. Moreover, we randomly repeated the genotyping for 10% of the whole sample and achieved 100% consistency. The manufacturer's software was used to analyze the results.

### 2.4. Haplotype Inference

We used the Bayesian statistical-based program PHASE (version 2.1; http://www.stat.washington.edu/stephens/software.html (accessed on 25 January 2021) to estimate haplotype frequencies. The possible haplotypes formed by the rs920829, rs16937976, and rs13218757 TRPA1 SNPs were CCG, CGA, CGG, CCA, TCG, and TGA. The CGG, CCA, and TGA haplotypes were observed in frequencies <2% and therefore were not included in subsequent analyses. The genotypes and haplotype frequencies are demonstrated in Table S1.

### 2.5. Statistical Analysis

Data were analyzed using Prism (GraphPad Software, La Jolla, CA, USA) and are presented as means ± SDs. Statistical significance was assessed with one-way ANOVA, with treatment interactions assessed by Tukey's post hoc multiple comparisons test. The sample sizes were adjusted after initial data collection. The study was exploratory research and did not test a prespecified statistical null hypothesis; therefore, the *p*-values are descriptive.

Continuous data were tested for normality. Clinical and laboratory characteristics of the patients are expressed as means ± SDs. In contrast, the changes in hemodynamic and biochemical parameters after the induction of anesthesia with propofol are expressed as means ± SEMs. A chi-squared test evaluated deviation from the Hardy–Weinberg equilibrium. The effects of TRPA1 genetic markers on hemodynamic and biochemical parameters were tested by analysis of variance followed by Tukey's posttest or Kruskal–Wallis followed by Dunn's posttest. When appropriate, Student's *t*-test or Mann–Whitney test was used according to the number of groups and parametric or nonparametric distributions of quantitative data. A $p < 0.05$ was considered statistically significant in all analyses. The power analysis calculations determined the minimum effects we could detect with our sample size.

We performed multiple linear regression (MLR) analyses to assess the contribution of Angiotensin-converting enzyme inhibitors (ACEi) used for pressure estimate (PE) separately. Crude regression coefficients (β) and 95% confidence intervals (CI) of possible predictors were respectively estimated with linear regression (model 1). In addition, we minimally adjusted for baseline characteristics (age and BMI) in model 2 and fully adjusted for baseline characteristics (age and BMI) and clinical factors (basal blood pressure) in model 3. Those covariates were adjusted as mediators for the association. In mutually adjusted models, we simultaneously included explanatory variables and polymorphisms (model 4). For this, we used a combination of the TC genotypes of TRPA1 rs920829 C>T SNP, CG+GG genotypes of TRPA1 rs16937976 C>G SNP, and of the AG+AA genotypes of TRPA1 rs13218757 G>A SNP, which correspond to the genotypes with the variant allele according to dbSNP. All *p* values were two-sided, and the alpha value was set at 0.05. Data were analyzed using Jamovi (Version 1.8), retrieved from https://www.jamovi.org (accessed on 10 October 2021).

## 3. Results

Table 1 shows the clinical and laboratory characteristics of the 164 patients enrolled in this study. There were decreases in SBP, MAP, and DBP ($p < 0.001$) after propofol administration. There was no difference in heart rate.

**Table 1.** Clinical and laboratory characteristics of patients.

| Characteristics | Total ($n$ = 164) |
|---|---|
| Male/female | 70/94 |
| Age (years) (mean $\pm$ SD) | 55 $\pm$ 14 |
| Ethnicity (Caucasian Yes/No) | 134/30 |
| Body mass index (kg/m$^2$) (mean $\pm$ SD) | 25.9 $\pm$ 4.7 |
| Total cholesterol (mg/dL (mean $\pm$ SD) | 192 $\pm$ 48 |
| Glucose (mg/dL) (mean $\pm$ SD) | 96 $\pm$ 19 |
| Urea (mg/dL) (mean $\pm$ SD) | 32 $\pm$ 12 |
| Creatinine (mg/dl) (mean $\pm$ SD) | 0.90 $\pm$ 0.3 |
| Potassium (mM) (mean $\pm$ SD) | 4.3 $\pm$ 0.5 |
| Hemoglobin (g/dL) (mean $\pm$ SD) | 12.7$\pm$ 1.9 |
| Total propofol use (mg) (mean $\pm$ SD) | 192 $\pm$ 68 |
| Mean blood pressure (mmHg) (mean $\pm$ SD) | |
| Baseline | 93.7 $\pm$ 16.5 |
| After propofol | 75.6 $\pm$ 15.8 * |
| Systolic blood pressure (mmHg) (mean $\pm$ SD) | |
| Baseline | 133 $\pm$ 19.7 |
| After propofol | 105 $\pm$ 18.2 * |
| Diastolic blood pressure (mmHg) (mean $\pm$ SD) | |
| Baseline | 76.8 $\pm$ 15.6 |
| After propofol | 64.1 $\pm$ 15.6 * |
| Heart rate (beats/min) (mean $\pm$ SD) | |
| Baseline | 81.4 $\pm$ 13.8 |
| After propofol | 79.4 $\pm$ 14.1 |

D: standard deviation; * $p < 0.001$ vs. baseline.

We evaluated the effects of TRPA1 genotypes on changes in SBP, MBP, and DBP after propofol administration. The different genotypes for rs920829 did not impact blood pressure changes (Figure 1A; $p > 0.05$). Interestingly, patients carrying the CG+GG genotypes for the rs16937976 polymorphism showed lower decreases in MBP and SBP in response to propofol than those carrying the CC genotype (Figure 1B; $p < 0.05$). In addition, we found more intense decreases in MBP and SBP in patients carrying the GG genotype for the rs13218757 polymorphism compared with patients carrying the AG+AA genotype (Figure 1C; $p < 0.05$).

To further investigate how variations in the *TRPA1* gene affect the hypotensive responses to propofol, we evaluated haplotypes formed by the TRPA1 rs920829, rs16937976, and rs13218757 SNPs. The least significant drop in mean and diastolic blood pressure values appeared for the CGA haplotype (Figure 2).

The MLR analyses assessing the role of ACEI use, age, BMI, basal blood pressure, and genotype in predicting changes in blood pressure promoted by propofol are presented below (Tables 2–4).

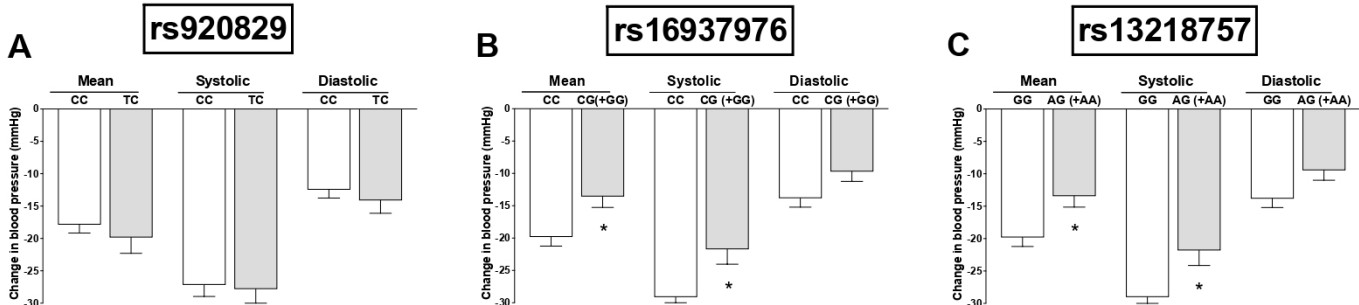

**Figure 1.** Effects of TRPA1 genotypes on blood pressure (BP) changes in response to propofol. (**A**) Effects of the CC (*n* = 127) and TC (*n* = 35) genotypes for the rs920829 polymorphism on changes in systolic, mean, and diastolic BP in response to propofol. (**B**) Effects of the CC (*n* = 119) and CG+GG (*n* = 44) genotypes for the rs16937976 polymorphism on changes in systolic, mean, and diastolic BP in response to propofol. (**C**) Effects of the GG (*n* = 120) and AG+AA (*n* = 42) genotypes for the rs13218757 polymorphism on changes in systolic, mean, and diastolic BP in response to propofol. Data are shown as means ± SEMs. * $p < 0.05$.

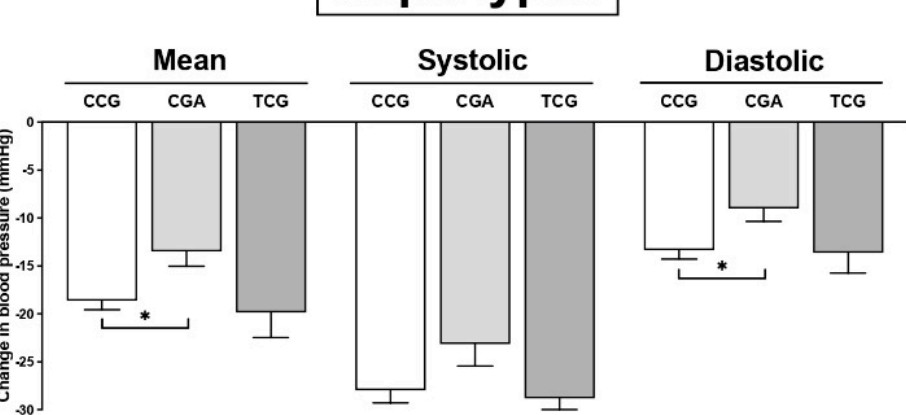

**Figure 2.** Effects of TRPA1 haplotypes on changes in systolic, mean, and diastolic blood pressure (BP) in response to propofol. CCG (*n* = 240), CGA (*n* = 46), and TCG (*n* = 31). Data are shown as means ± SEMs. * $p < 0.05$.

**Table 2.** Effect of rs920829 polymorphism in *TRPA1* gene on changes in blood pressure induced by propofol after adjustment for selected variables.

| | Change in SBP (mmHg) | | | Change in MBP (mmHg) | | | Change in DBP (mmHg) | | |
|---|---|---|---|---|---|---|---|---|---|
| Rs920829 | $R^2 = 0.40$ | RMSE = 15.00 | | $R^2 = 0.28$ | RMSE = 12.20 | | $R^2 = 0.23$ | RMSE = 11.90 | |
| Source | ß | 95% CI | *p* | ß | 95% CI | *p* | ß | 95% CI | *p* |
| Use of ACEi | +1.64 | −6.65 to 9.94 | 0.695 | −2.18 | −8.84 to 4.47 | 0.518 | −1.36 | −7.87 to 5.15 | 0.680 |
| Age (years) | −0.02 | −0.24 to 0.19 | 0.804 | −0.04 | −0.21 to 0.13 | 0.636 | +0.00 | −0.15 to 0.17 | 0.943 |
| BMI (kg/m$^2$) | +0.58 | −0.05 to 1.21 | 0.071 | +0.72 | 0.21 to 1.24 | <0.006 * | +0.67 | 0.17 to 1.18 | 0.009 * |
| BBP | −0.61 | −0.76 to −0.47 | <0.001 * | −0.43 | −0.56 to −0.31 | <0.001 * | −0.40 | −0.53 to −0.27 | <0.001 * |
| TC | −3.61 | −10.09 to 2.65 | 0.251 | −2.65 | −7.83 to 2.51 | 0.311 | −2.77 | −7.83 to 2.29 | 0.281 |

Abbreviations: ACEi-Angiotensin-converting enzyme inhibitors; BMI-body mass index; β-parameter estimate; CI-confidence interval; BBP-basal blood pressure. Reference genotype: CC. * $p < 0.05$.

**Table 3.** Effect of rs16937976 polymorphism in *TRPA1* gene on changes in blood pressure induced by propofol after adjustment for selected variables.

| | Change in SBP (mmHg) | | | Change in MBP (mmHg) | | | Change in DBP (mmHg) | | |
|---|---|---|---|---|---|---|---|---|---|
| **Rs16937976** | $R^2 = 0.42$ | RMSE = 15.40 | | $R^2 = 0.32$ | RMSE = 11.80 | | $R^2 = 0.26$ | RMSE = 11.70 | |
| **Source** | ß | 95% CI | *p* | ß | 95% CI | *p* | ß | 95% CI | *p* |
| Use of ACEi | +0.97 | −7.49 to 9.44 | 0.820 | −1.98 | −8.43 to 4.45 | 0.543 | −1.20 | −7.60 to 5.18 | 0.709 |
| Age (years) | −0.11 | −0.33 to 0.11 | 0.318 | −0.04 | −0.20 to 0.12 | 0.612 | +0.00 | −0.15 to 0.16 | 0.954 |
| BMI (kg/m$^2$) | +1.20 | 0.55 to 1.85 | <0.001 * | +0.82 | 0.32 to 1.13 | 0.001 * | +0.75 | 0.25 to 1.25 | 0.003 * |
| BBP | −0.62 | −0.77 to −0.47 | <0.001 * | −0.44 | −0.56 to −0.32 | <0.001 * | −0.40 | −0.53 to −0.28 | <0.001 * |
| CG (+GG) | +7.53 | 1.62 to 13.43 | 0.013 * | +7.15 | 2.61 to 11.70 | 0.002 * | +5.18 | 0.67 to 9.70 | 0.025 * |

Abbreviations: ACEi-Angiotensin-converting enzyme inhibitors; BMI-body mass index; β-parameter estimate; CI-confidence interval; BBP-basal blood pressure. Reference genotype: CC. * $p < 0.05$.

**Table 4.** Effect of rs13218757 polymorphism in *TRPA1* gene on changes in blood pressure induced by propofol after adjustment for selected variables.

| | Change in SBP (mmHg) | | | Change in MBP (mmHg) | | | Change in DBP (mmHg) | | |
|---|---|---|---|---|---|---|---|---|---|
| **Rs13218757** | $R^2 = 0.42$ | RMSE = 15.30 | | $R^2 = 0.33$ | RMSE = 11.70 | | $R^2 = 0.26$ | RMSE = 11.70 | |
| **Source** | ß | 95% CI | *p* | ß | 95% CI | *p* | ß | 95% CI | *p* |
| Use of ACEi | +0.82 | −7.62 to 9.26 | 0.848 | −2.13 | −8.55 to 4.27 | 0.511 | −1.31 | −7.70 to 5.07 | 0.685 |
| Age (years) | −0.10 | −0.32 to 0.11 | 0.339 | −0.03 | −0.20 to 0.12 | 0.637 | +0.00 | −0.15 to 0.16 | 0.932 |
| BMI (kg/m$^2$) | +1.24 | 0.59 to 1.89 | <0.001 * | +0.86 | 0.36 to 1.36 | <0.001 * | +0.77 | 0.28 to 1.27 | 0.002 * |
| BBP | −0.62 | −0.77 to −0.47 | <0.001 * | −0.44 | −0.56 to −0.31 | <0.001 * | −0.40 | −0.53 to −0.27 | <0.001 * |
| AG (+AA) | +8.11 | 2.12 to 14.10 | 0.008 * | +7.66 | 3.06 to 12.27 | 0.001 * | +5.62 | 1.04 to 10.21 | 0.016 * |

Abbreviations: ACEi—Angiotensin-converting enzyme inhibitors; BMI—body mass index; β—parameter estimate; CI—confidence interval; BBP—basal blood pressure. Reference genotype: GG. * $p < 0.05$.

Our second question was whether TRPA1 genotypes and haplotypes influence the changes in plasma NOx, nitrate, and nitrite levels in response to propofol anesthesia. Patients carrying the TC genotype for the rs920829 polymorphism showed higher increases in NOx and nitrate levels in response to propofol than carriers of the CC genotype (Figure 3A; $p < 0.05$), without any significant difference in the blood pressure. On the other hand, we discovered that the rs16937976 and rs13218757 polymorphisms and CGA haplotype impacted blood pressure decrease without affecting NOx, nitrite, or nitrate levels after propofol anesthesia (Figure 3B,C; $p > 0.05$). None of the studied haplotypes showed a significant change for NOx or nitrate variations. ($p > 0.05$; data not shown).

Regarding the haplotypes, after linear regression analysis, we also found a significant difference in the systolic pressure (Table 5; * $p < 0.05$) for the CGA haplotype compared with CGG.

None of the studied polymorphisms or haplotypes showed a significant change in heart rate values (Tables S2 and S3). TRPA1 genotypes and haplotypes did not affect the baseline values (prior to propofol injection) of SBP, MBP, DBP, or HR (Supplementary Figure S1, $p > 0.05$).

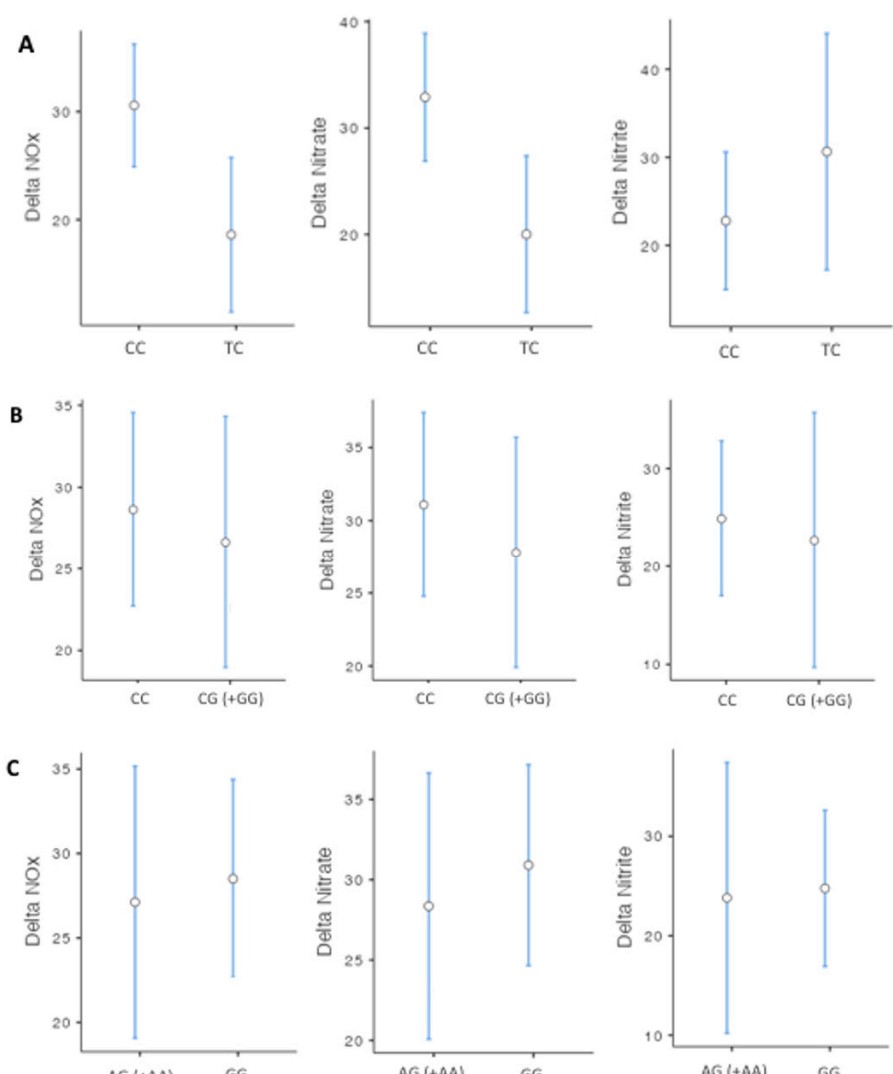

**Figure 3.** Impact of TRPA1 genotypes on alterations in NOx, nitrite, and nitrate concentrations induced by propofol. (**A**) Effects of the CC (*n* = 127) and CG (*n* = 35) genotypes for the rs920829 SNP. (**B**) Effects of the CC (*n* = 119) and CG+GG (*n* = 44) genotypes for the rs16937976 SNP. (**C**) Effects of the GG (*n* = 120) and AG+AA (*n* = 42) genotypes for the rs13218757 SNP. Data are shown as means ± SDs.

**Table 5.** Effect of *TRPA1* haplotypes on changes in blood pressure induced by propofol after adjustment for selected variables.

| | Change in SBP (mmHg) | | | Change in MBP (mmHg) | | | Change in DBP (mmHg) | | |
|---|---|---|---|---|---|---|---|---|---|
| | $R^2$ = 0.39 | RMSE = 15.40 | | $R^2$ = 0.28 | RMSE = 11.70 | | $R^2$ = 0.24 | RMSE = 11.70 | |
| **Source** | ß | 95% CI | *p* | ß | 95% CI | *p* | ß | 95% CI | *p* |
| Use of ACEi | −0.74 | −4.78 to 6.27 | 0.790 | −1.46 | −5.64 to 2.72 | 0.493 | +0.10 | −4.06 to 4.26 | 0.962 |
| Age (years) | −0.09 | −0.23 to 0.05 | 0.207 | −0.07 | −0.17 to 0.03 | 0.199 | −0.00 | −0.11 to 0.09 | 0.877 |
| BMI (kg/m²) | +0.98 | +0.56 to 1.41 | <0.001 * | +0.72 | 0.39 to 1.04 | <0.001 * | +0.63 | 0.31 to 0.95 | <0.001 * |
| BBP | −0.60 | −0.70 to −0.50 | <0.001 * | −0.40 | −0.49 to −0.32 | <0.001 * | −0.38 | −0.47 to 0.30 | <0.001 * |
| | | | | Haplotypes | | | | | |
| CGA | +6.04 | 0.90 to 11.08 | 0.021 * | +5.74 | 1.81 to 9.67 | 0.004 * | +4.55 | 0.65 to 8.45 | 0.022 * |
| CGG | +3.73 | −27.16 to 34.63 | 0.812 | +2.02 | −21.58 to 25.62 | 0.866 | −0.04 | −23.52 to 23.42 | 0.997 |
| TCG | +0.41 | −5.58 to 6.41 | 0.892 | −1.27 | −5.854 to 3.31 | 0.586 | −1.90 | −5.64 to 3.45 | 0.636 |

Abbreviations: ACEi—Angiotensin-converting enzyme inhibitors; BMI—body mass index; β—parameter estimate; CI—confidence interval; BBP—basal blood pressure. Reference haplotype: CGG * *p* < 0.05.

## 4. Discussion

This study was a pioneer in evaluating whether TRPA1 genotypes and haplotypes affect hemodynamic parameters and NO bioavailability after propofol anesthesia. The main findings were that: (i) variant genotypes (CG+GG) for the rs16937976 polymorphism in the *TRPA1* gene were associated with lower blood pressure drops after propofol anesthesia compared to the ancestral genotype CC; (ii) ancestral genotype (CC genotype) for the rs13218757 polymorphism in the *TRPA1* gene was associated with more intense decreases in blood pressure after propofol anesthesia compared with variant (AG+AA) genotypes; (iii) the TRPA1 CGA haplotype was associated with lower decreases in blood pressure after propofol anesthesia compared with the CGG haplotype; (iv) patients carrying the CT genotype for the rs920829 polymorphism showed a higher increase in NOx and nitrate levels in response to propofol than carriers of the CC genotype and rs16937976 and rs13218757 polymorphisms, and the haplotypes did not affect the changes in NOx, nitrite, or nitrate levels after propofol anesthesia. None of the studied genotypes or haplotypes showed a significant change in HR variation.

Pereira et al. [28] discovered that TRPA1 inhibition reduces visceral nociception through a mechanism unrelated to the modulation of resident cells, nitric oxide, and opioid pathways. Similarly, our findings suggest that some TRPA1 polymorphisms and haplotypes may reduce propofol hypotension response by a mechanism independent of the nitric oxide pathways.

Numerous studies have identified that propofol has vasodilatory properties in vivo [28] and in vitro [6,29,30]. Moreover, in sensory neurons and heterologous expression systems, propofol has been demonstrated to activate and modify TRPA1 ion channel sensitivity to agonist activation [13,31]. However, a link between anesthetics and TRPA1 activation in modulating vasomotor tone in vivo has yet to be established. Sinha et al. [19] described that propofol-induced depressor responses in vivo are mediated partly by TRPA1 channels. Moreover, combined inhibition of eNOS or BKCa channels virtually abolishes the vasodilator effect, whereas individual inhibition markedly attenuates the propofol-induced depressor response. Still, according to Sinha et al., the effect of eNOS and BKCa inhibition is lost in TRPA1 knockout (KO) mice. More recently, Talavera et al. [18] clearly described two mechanisms that have been proposed to explain TRPA1-dependent vasodilation: nerve-evoked vasodilation and endothelium-dependent vasodilation. The activation of the TRPA1 channel in perivascular sensory nerves leads to a $Ca^{2+}$ influx that subsequently causes the release of calcitonin gene-related peptide (CGRP) from the sensory nerves that innervate the vascular wall. CGRP binds its G protein-coupled receptor (GPCR) on smooth muscle cells (SMCs) and causes membrane hyperpolarization, myocyte relaxation, and arterial dilation. TRPA1 agonists in the bloodstream may activate the channel in endothelial cells, resulting in $Ca^{2+}$ influx. High intracellular $Ca^{2+}$ leads to hyperpolarization of the endothelial cells via the stimulation of the $Ca^{2+}$-activated K+ channels (KCa).

The mechanism through which propofol acts on the vasculature is controversial. It may involve direct modulation of vascular tone in an endothelium-dependent or endothelium-independent manner depending on the species (rat, pig, and human) and vascular bed (thoracic and coronary) from which the arterioles were obtained [18]. Specifically, endothelial denuding [30,32] and e-NOS inhibition [6] resulted in sustained dilation by propofol, whereas Klockgether et al. [5] demonstrated a role for BK channels in the response.

TRPA1 channels have been identified as essential modulators of vasomotor tone in vivo and in vitro [22,33]. Sinha et al. [19] observed a dose-dependent decrease in MAP following the administration of clinically relevant propofol concentrations in control mice. Propofol-induced decrease in MAP was markedly attenuated (>50%) in TRPA1 KO mice. The depressor response to propofol, at the lowest concentration of propofol [2.5 mg/kg] tested, in the TRPA1 KO mice was virtually abolished. This dosage [2–2.5 mg/kg] represents a clinically relevant dose typically used for the induction of anesthesia. Pozsgai et al. [22] also suggested that TRPA1 may influence changes in blood pres-

sure of possible relevance to autonomic system reflexes and potentially to vasovagal/neuro-cardiogenic syncope disorders.

The classical mechanism proposed for the propofol vasorelaxant effect is through NO upregulation [19,34]. Indeed, propofol promotes dose-dependent hypotensive effects [19,34–36], which are at least in part related to increases in eNOS-derived NO formation [19,34]. Interestingly, some studies have suggested that propofol activates eNOS, possibly by inducing phosphorylation of Ser1177, thereby leading to enzyme activation and NO synthesis [37,38]. Ex vivo and in vivo studies have shown that eNOS is involved in the cardiovascular effects of propofol, which is consistent with this mechanism [19,34,39]. Aligned with this evidence and our previous reports [2,8], we observed increased plasma nitrite concentrations after propofol administration, suggesting an increased NO bioavailability induced by this anesthetic.

Regarding the interference of the studied polymorphisms, we found that NOx and nitrate levels were affected by the rs920829 polymorphism. Patients with genotype CC showed higher increases in NOx and nitrate levels in response to propofol than carriers of the CT genotype. However, this was insufficient to produce more intense hypotension and affect the nitrite levels after propofol infusion for unknown reasons. Indeed, NO has a very brief half-life, limiting the evaluation of endogenous NO production in vivo [40]. Therefore, the measurement of NO oxidation products (nitrite and nitrate) is commonly used as an index of NO bioavailability [26,41]. While the assessment of nitrate in plasma has frequently been used as a parameter of NO formation, several studies have consistently demonstrated that measuring circulating nitrite concentrations results in much better information [42–44]. The clinical utility of assessing nitrate to evaluate NO formation may be limited by many interfering factors such as diet, clinical conditions, medications, smoking status, and other environmental factors [41]. Therefore, the possible effects of TRPA1 variants on NO signaling may not be reflected by plasma nitrite levels.

The rs16937976 and rs13218757 polymorphisms and the haplotypes did not present any additional impact on the NOx, nitrite, or nitrate levels, leading us to conclude that these TRPA1 polymorphisms may modify blood pressure decrease induced by propofol throughout another mechanism than modulating NO bioavailability.

Propofol is a TRPA1 agonist on sensory neurons, as identified by Lee et al. [45]. TRPA1 activation mediates the decrease in mean arterial pressure and dilatation of murine coronary microvessels induced by propofol via a mechanism involving the activation of eNOS as well as BKCa channels [19,46], restoring the sensitivity of TRPV1 via eNOS-dependent activation of protein kinase C-$\varepsilon$ (PK C-$\varepsilon$) [47]. Recent structure–function, molecular modeling, and photoaffinity labeling studies strongly suggest that propofol binds TRPA1 at various sites [11,12].

The application of TRPA1 agonists causes the dilation of several arteries, and these responses are minor if vessels are treated with TRPA1 blockers or in preparations isolated from TRPA1-deficient mice [33,48,49]. Notably, TRPA1-mediated vasodilation in mouse mesenteric artery rings occurs in an endothelial and neuropeptide-independent manner [22]. These findings raise the possibility that not only mechanisms mediated by sensory nerves are responsible for TRPA1-dependent vasorelaxation.

Moreover, the depressor response provoked by propofol is not entirely eliminated when TRPA1 is deleted [19]. Even combined inhibition of eNOS and BKCa channels eliminates the effect, suggesting that propofol directly interacts with eNOS, BKCa channels, or both, in vivo [47].

Alternatively, other mediators and channels may be involved. Some previous reports have demonstrated a role for ATP-gated K+ channels and cyclooxygenase products in mediating propofol-induced depressor responses [50,51]. Thus, rs16937976 and rs13218757 polymorphisms and haplotypes possibly modified these mechanisms, leading to a lower blood pressure after propofol infusion.

Furthermore, TRPA1 agonists cause the release of adrenaline from the adrenal cortex [52], leading to systemic cardiovascular changes and a variety of vascular actions in

distinct vascular beds. A proposed cooperative action of NO is required for the TRPA1-CGRP signaling pathway and the regulation of the vascular tone [53,54]. Intriguingly, mesenteric arterioles from TRPA1 KO mice relax significantly less in response to NO than arteries from wild-type animals [55]. Remarkably, the relaxation of the blood vessel wall is accompanied by increased CGRP and NO levels. It is suggested that TRPA1-expressing sensory neurons may be involved in the vascular component of neurogenic inflammation. Moreover, TRPA1 expression increases in macrophage-foam cells in mouse atherosclerotic aortas [24]. As aging is often accompanied by arterial atherosclerotic appearance, this may explain the correlation between age and blood pressure decrease in our study.

Patients undergoing colonoscopy have risk factors for hypotension, such as bowel preparation, age, and comorbidities, and are thus an appropriate group to monitor for sedation-induced hypotension [56]. According to some researchers, hypotension during induction is more common in patients who have received ACE inhibitors up to the day of surgery [57,58]. In the present study, we found no independent associations between preoperative ACEi therapy and higher post-propofol hypotension, probably because all patients were advised to refrain from taking these drugs for 24 h before the procedure [59]. Additionally, following previous findings, baseline BP level was an important predictor of BP response to propofol infusion [60]. Age was not correlated with higher rates of hypotension. We attribute this to the fact that the individuals in the study were, for the most part, relatively young. Kawasaki et al. showed that age of 65 years or older was a significant predictive factor for hypotension caused by propofol injection [61].

There are some limitations in our study. We did not examine other polymorphisms in genes encoding proteins related to TRPA1 modulation that could affect the hypotensive responses to propofol, such as PRKCA [61,62] and VEGF [63]. Therefore, other genetic polymorphisms remain to be evaluated.

In summary, our results present that propofol anesthesia induces more intense hypotensive responses in patients with dominant homozygous genotypes (CC and GG) to *TRPA1* gene polymorphisms rs16937976 and rs13218757. Considering the haplotypes, the most negligible variations in systolic, mean, and diastolic blood pressure drops were for the CGA haplotype. Interestingly, there was no relationship between higher baseline blood pressure values for any genotypes in the studied polymorphisms. Moreover, they did not correlate in prevalence regarding age, BMI, or sex among the studied patients.

With improved genetic knowledge, it is expected that medications will be better indicated for each patient based on their genotype and haplotype in the future. Furthermore, our findings could help predict the hemodynamic side effects of propofol, a common anesthetic. Finally, because anesthetics activate TRPA1, it could help with drug development, particularly when it comes to modifying propofol management in uncompensated hypertensive and hypotensive patients.

**Supplementary Materials:** The following supporting information can be downloaded at: https://www.mdpi.com/article/10.3390/cimb44120432/s1, Figure S1: Effects of TRPA1 genotypes and haplotypes on blood pressure (BP) baseline values (before propofol injection). (A) Effects of the CC (*n* = 127) and TC (*n* = 35) genotypes for the rs920829 polymorphism on systolic, mean, and diastolic BP. (B) Effects of the CC (*n* = 119) and CG+GG (*n* = 44) genotypes for the rs16937976 polymorphism on systolic, mean, and diastolic BP. (C) Effects of the GG (*n* = 120) and AG+AA (*n* = 42) genotypes for the rs13218757 polymorphism on systolic, mean, and diastolic BP. (D) Effects of the haplotypes on systolic, mean, and diastolic BP (Data are shown as means ± SEMs. *p* > 0.05); Table S1: Genotype and Haplotypes frequencies for *TRPA1* polymorphisms; Table S2: Effect of *TRPA1* genotypes on changes in heart rate induced by propofol after adjustment for selected variables; Table S3: Effect of *TRPA1* haplotypes on changes in heart rate induced by propofol after adjustment for selected variables.

**Author Contributions:** Conceptualization, W.N.P.-G., R.L., G.H.O.-P. and J.E.T.-S.; Methodology, I.B.d.M., W.N.P.-G., G.H.O.-P., L.P.F., R.L., G.C.F., L.C.P. and L.V.G.; Writing—Original Draft Preparation, W.N.P.-G. and I.B.d.M.; Writing—Review and Editing, W.N.P.-G., G.H.O.-P. and R.L.; Supervision, W.N.P.-G., J.E.T.-S., G.H.O.-P. and R.L.; Project Administration, W.N.P.-G., G.H.O.-P. and J.E.T.-S.; Funding Acquisition, W.N.P.-G., J.E.T.-S. and R.L. All authors have read and agreed to the published version of the manuscript.

**Funding:** This research received no external funding.

**Institutional Review Board Statement:** This study was conducted following the Declaration of Helsinki, and approval for the use of human subjects was obtained from the Institutional Review Board at the Ribeirao Preto Medical School, University of Sao Paulo, Brazil. The study is registered at ClinicalTrials.gov (identifier NCT02442232).

**Informed Consent Statement:** Informed consent was obtained from all subjects involved in the study.

**Data Availability Statement:** Not applicable.

**Acknowledgments:** Fundação de Apoio ao Ensino, Pesquisa e Assistência do Hospital das Clínicas da Faculdade de Medicina de Ribeirão Preto da Universidade de São Paulo (FAEPA).

**Conflicts of Interest:** The authors declare no conflict of interest.

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
