# Peer review of "TRPA1 Polymorphisms Modify the Hypotensive Responses to Propofol with No Change in Nitrite or Nitrate Levels"

_cimb, doi:10.3390/cimb44120432_

Round 1

Reviewer 1 Report

Borges de Melo and colleagues investigated the association between TRPA1 polymorphisms and the hypotensive responses to propofol. Although the rationale is well described and the data are clearly presented, some improvements are required:

Major points: The authors should discuss or speculate on the correlative analysis they provide in the tables. They are not even commented in the discussion. Furthermore, the authors should put their findings in a clinical context: Why is it important to consider TRPA1 polymorphisms? None of the blood pressure declines was critical. Might the situation change if severely hypotensive patients had been included in the study? Could we probably use the polymorphism knowledge (easy to measure) to modify propofol management in hypotensive patients. As the manuscript is presented, it gives the impression of some loosely related measurements, which are based on animal and in vitro studies as background.

Minor comments:

Line 81: please define predetermined time.

Line 90: Please explain how you determine NO marker levels at 4 and 10 minutes when you collect blood samples only at 10 minutes.

Line 114: Table S3 is mentioned before all other data, should be renumbered as S1.

Line 150: Incomplete sentence

Table 1: Ethnicity is incorrect and incomplete (Caucasian + ?)

Tables: please explain abbreviations in the Footnotes

The Figure S2 is not in the submission. Anyway, this information should be depicted in the main manuscript.

Line 235, expressive – wrong use

Author Response

Dear Reviewer,

We want to thank you for the exciting and insightful comments and the opportunity to improve our manuscript. Please find below our responses to these comments. Changes in the revised manuscript are indicated in red font.

Reviewer 1

Borges de Melo and colleagues investigated the association between TRPA1 polymorphisms and the hypotensive responses to propofol. Although the rationale is well described and the data are clearly presented, some improvements are required:

Major points: The authors should discuss or speculate on the correlative analysis they provide in the tables. They are not even commented in the discussion. Furthermore, the authors should put their findings in a clinical context: Why is it important to consider TRPA1 polymorphisms? None of the blood pressure declines was critical. Might the situation change if severely hypotensive patients had been included in the study? Could we probably use the polymorphism knowledge (easy to measure) to modify propofol management in hypotensive patients. As the manuscript is presented, it gives the impression of some loosely related measurements, which are based on animal and in vitro studies as background.

-Response to the Reviewer:

We appreciate such constructive feedback. We addressed the table results in the discussion section and added more information to clarify the potential clinical benefit of better polymorphism knowledge.

Minor comments:

-Line 81: please define predetermined time.

-Response to the Reviewer: We added the precise times at the sentence.

Line 90: Please explain how you determine NO marker levels at 4 and 10 minutes when you collect blood samples only at 10 minutes.

-Response to the Reviewer: Thank you for bringing this poorly worded sentence to our attention. We reworded the text to make it clearer.

Line 114: Table S3 is mentioned before all other data, should be renumbered as S1.

-Response to the Reviewer: The supplementary tables are now numbered correctly.

Line 150: Incomplete sentence

-Response to the Reviewer: We fixed the sentence.

Table 1: Ethnicity is incorrect and incomplete (Caucasian + ?)

-Response to the Reviewer: Thank you for noticing the incomplete information. We corrected the table.

Tables: please explain abbreviations in the Footnotes

-Response to the Reviewer: We described the abbreviations in the Footnotes of all tables.

The Figure S2 is not in the submission. Anyway, this information should be depicted in the main manuscript.

-Response to the Reviewer: We appreciate and accept the excellent suggestion of depicting this information in the main manuscript. This figure is now number 3.

Line 235, expressive – wrong use

-Response to the Reviewer: We removed the word from the sentence.

Reviewer 2 Report

I have read this paper with great interest, and value the effort, but we need more info on the methods (including the study registration), and the clinical 'impact (decrease versus absolute values)

We need some more detailed information on how blood pressure data were collected (materials, etc) as this is a crucial finding for the study.
I assume that the clinicians involved in data acquisition were blinded for the polymorphisms of interest ?

These are not volunteers, but patients

Table 1, were data truly normally distributed ? I expected a skewed pattern.

It is intriguing to notice that mean and diastolic changes are more pronounced compared to systolic findings. However, can the changes be modulated by the pre-sedation findings (as changes in blood pressure are reported). Are pre-exposure blood pressures modulated by the polymorphisms involved Related to this, are absolute values different between the different haplotypes, or not, as this could be relevant for the clinical relevance ?

When reading the protocol as published at clinicaltrials.gov, the research question and approach taken reads quite different ? is this a secondary analysis, or how should I understand this (cf ref 33) ?

Author Response

Dear Reviewer,

We want to thank you for the exciting and insightful comments and the opportunity to improve our manuscript. Please find below our responses to these comments. Changes in the revised manuscript are indicated in red font.

Reviewer 2

I have read this paper with great interest, and value the effort, but we need more info on the methods (including the study registration), and the clinical 'impact (decrease versus absolute values)

We need some more detailed information on how blood pressure data were collected (materials, etc) as this is a crucial finding for the study.
I assume that the clinicians involved in data acquisition were blinded for the polymorphisms of interest?

-Response to the Reviewer: Thank you. Yes, the clinicians involved in data collection were blinded to the polymorphisms of interest. We added more details at Methods´s section.

These are not volunteers, but patients

-Response to the Reviewer: Thank you. We replaced volunteers by patients.

It is intriguing to notice that mean and diastolic changes are more pronounced compared to systolic findings. However, can the changes be modulated by the pre-sedation findings (as changes in blood pressure are reported). Are pre-exposure blood pressures modulated by the polymorphisms involved Related to this, are absolute values different between the different haplotypes, or not, as this could be relevant for the clinical relevance?

-Response to the Reviewer: We also agree that it is very intriguing this finding. However, the TRPA1 genotypes and haplotypes did not affect the baseline values (prior to propofol injection) of SBP, MBP, DBP and HR (Supplementary Figure S1, p>0.05).

When reading the protocol as published at clinicaltrials.gov, the research question and approach taken reads quite different? is this a secondary analysis, or how should I understand this (cf ref 33)?

-Response to the Reviewer: Yes, It is a secondary analysis. The purpose was also to evaluate if some other polymorphisms can impact the hemodynamic responses to propofol and the effects of these polymorphisms on biomarkers related to nitric oxide after propofol anesthesia.

Reviewer 3 Report

Thank you for the possibility to review the manuscript titled: “TRPA1 polymorphisms modify the hypotensive responses to propofol with no change in nitrite or nitrate levels”. The manuscript is ell written and easy to read. It also provides important insights in our understanding of the mechanisms of hypotension in patients who receive propofol. There are no major objections, only several minor corrections:

-In the first table n=164 should be moved up in the Parameters/Data section. Moreover, please be more specific and include a description in the table that the data is presented as mean±standard deviation (table 1)

Please take into account the recommendations in the spirit of improving the quality of the submission.

Author Response

Dear Reviewer,

We want to thank you for the exciting and insightful comments and the opportunity to improve our manuscript. Please find below our responses to these comments. Changes in the revised manuscript are indicated in red font.

Reviewer 3

Thank you for the possibility to review the manuscript titled: “TRPA1 polymorphisms modify the hypotensive responses to propofol with no change in nitrite or nitrate levels”. The manuscript is ell written and easy to read. It also provides important insights in our understanding of the mechanisms of hypotension in patients who receive propofol. There are no major objections, only several minor corrections:

-In the first table n=164 should be moved up in the Parameters/Data section. Moreover, please be more specific and include a description in the table that the data is presented as mean±standard deviation (table 1)

-Response to the Reviewer: Thank you very much for reviewing this manuscript! We edited table 1 following the suggestions.

Round 2

Reviewer 1 Report

In the newly written paragraph, citations should be included in the sentence, not after. This is a minor remaining error, which could be also fixed during editing. Otherwise, the authors responded to the criticisms.

Author Response

Reviewer 1

In the newly written paragraph, citations should be included in the sentence, not after. This is a minor remaining error, which could be also fixed during editing. Otherwise, the authors responded to the criticisms.

-Response to the Reviewer:

Thank you for noticing the inadequate citation. We corrected it in the paragraph.

Reviewer 2 Report

i only have one additional suggestion: the authors confirm that this is a secondary analysis, but this is not yet sufficiently clear in the paper, so that i recommend to make this clearer (abstract, methods)

Author Response

Reviewer 2

I only have one additional suggestion: the authors confirm that this is a secondary analysis, but this is not yet sufficiently clear in the paper, so that I recommend to make this clearer (abstract, methods)

-Response to the Reviewer:

We appreciate the suggestion and edited the manuscript accordingly (abstract and methods) to clarify this information.